# Mesothelioma Mortality Rates in Greece for the Period 2005–2015 Is Increased Compared to Previous Decades

**DOI:** 10.3390/medicina55080419

**Published:** 2019-07-30

**Authors:** Evdoxia Gogou, Chrissi Hatzoglou, Sotirios G. Zarogiannis, Foteini Malli, Rajesh M. Jagirdar, Konstantinos I. Gourgoulianis

**Affiliations:** 1Department of Physiology, Faculty of Medicine, University of Thessaly, BIOPOLIS, 41500 Larissa, Greece; 2Anatomy and Physiology Lab, Nursing Department, University of Thessaly, 41500 Larissa, Greece; 3Department of Respiratory Medicine, Faculty of Medicine, University of Thessaly, BIOPOLIS, 41500 Larissa, Greece

**Keywords:** asbestos, epidemiology, Greece, malignant mesothelioma, mortality rate

## Abstract

*Background and Objective:* To present summary statistics regarding malignant mesothelioma (MM) mortality in Greece during the period 2005–2015 and compare it with previous decades, along with gender, age and geographical area analysis. *Materials and Methods*: The Hellenic Statistical Authority provided the data, which included all deaths for the period 1983 to 2015 that mentioned MM as the death cause in the corresponding death certificate. MM mortality rates have been calculated with respect to gender, age, and geographical location in Greece. Furthermore, a comparison analysis was made among three eleven consecutive year periods, in order to assess potential changes in the mortality rates. *Results*: The MM mortality rate has significantly increased during the period 2005–2015 both in males and females compared to earlier decades. The maximum number of MM deaths has shifted to an older age group of 70–80 years during the 2005–2015 period as compared to that of 1983–2004 in both genders. Additionally, MM mortality rates have significantly increased in all geographical areas except for the Epirus Prefecture. *Conclusions*: Our results demonstrate an increased MM mortality rate in Greece for the decade 2005–2015 as compared to the two previous decades. This increase is possibly due to the fact that the peak in asbestos production and use in Greece was in mid 1990s, while the asbestos ban came in effect in 2005. Based on these findings the MM epidemic in Greece has not yet peaked, therefore it is important to implement screening strategies for early MM detection.

## 1. Introduction

Mesothelioma is a malignant tumor which arises from the mesothelial or sub mesothelial cells of the pleura, peritoneum, pericardium and tunica vaginalis [1,2]. More than 80% of all mesothelioma cases originate from the pleura and approximately 75–80% of patients are males [2,3]. Mesothelioma is officially recognized as an occupational cancer and the association between asbestos exposure and mesothelioma is well established [4]. Exposure to asbestos is considered as the primary aetiologic agent of mesothelioma. Other causes of malignant mesothelioma(MM) include radiation, chronic serosal inflammation and asbestos-like fibers (erionite, fluoro-edenite, nanotubes). Erionite is a mineral found in rock in certain areas of Turkey which is the cause of mesothelioma epidemic in Cappadocia [5]. Also, erionite exposure in Dunn County of North Dakota is similar to that of Cappadocia villages. Cases of MM in the Sicilian town of Biancavilla found to be related to the fluoro-edenite fibers [6]. Long CNTs (carbon nanotubes) or long asbestos fibers into the pleural cavity of mice induces mesothelioma [7]. Frequent mutations in BAP1 (BRCA-associated protein 1) have been reported in many types of cancer including uveal melanoma, renal cancer and MM. A recent study proposed that germline BAP1 mutations are linked with cancer syndrome which is associated with familiar cases of MM in the above-mentioned Cappadocian villages [5,8].

Two independent epidemiological studies conducted in Liguria, Italy, and Ontario, Canada have shown that 96–100% of the MMs and 42–49% of lung tumors arising among petroleum workers were attributable to asbestos exposure [9]. Whenever MM is considered as a possible diagnosis, thorough occupational history should be taken to cover all occupations throughout life and the patient should be referred to an experienced pulmonary department. In the context of proper clinical, radiological, and thoracoscopic/surgical findings the final diagnosis of MM is always histological (on biopsy or resection); in a few particular cases it can be made on cytological material [10].

Upon diagnosis, MM is rapidly progressive and invariable fatal disease [11]. The mean survival is 9–14 months post-diagnosis and despite growing research in the field, the mean survival has remained practically unchanged [12,13,14]. The mean latency period for MM development is 30–40 years, however longer periods of about 70years have also been reported [4,15].Because of the strong causative relationship between asbestos and MM, the incidence and mortality trends observed follow the pattern of asbestos exposure trends [4].According to standard epidemiology principles, in the case of a rare disease when the mean survival from diagnosis to death is about a year, then the incidence, the prevalence and the mortality rate do not differ [16].

Previous published data about MM mortality rate in Greece, during the period 1983–2003, have shown a remarkable three-fold increase on the cause-specific mortality rate during the decade 1994–2003 which was 0.156/100.000population, comparing with the rate 0.047/100.000population in a span from 1983 to 1993. Furthermore, the area of Epirus had the highest MM mortality rate over the period 1983–2003 which was 0.38/100.000population and the lower was 0.025/100.000 in Thessalía [17].

In Greece, the production of asbestos started at 1960s and this industry flourished in 1970 onwards, with a peak in production in the 1990s [18,19]. Taking into consideration the long latency period of MM, increased mortality rate of this disease is expected during the early decades of 21th century.

This study aimed at assessing the MM mortality rate in Greece for the period 2005–2015 for both sexes in different geographical areas of the country and to highlight potential differences with previously published data.

## 2. Materials and Methods

The study was based on data provided by the Hellenic Statistical Authority. These data contain all deaths from 1983 to 2015 where MM was mentioned as cause of death on a patient’s death certificate. The deaths attributed to MM where deaths in patients with a confirmed MM diagnosis and/or in patients where mesothelioma was defined as a cause of death at autopsy. Furthermore, data regarding the age, the sex and the residence were also mentioned. The primary aim of the study was to present summary statistics of mortality due to MM in Greece between 2005 and 2015, with comments relating to time trends, age, gender and geographical differences. The data were provided by the Hellenic Statistical Authority with full terms of confidentiality. In accordance with the Hellenic Statistical Authority, no ethical committee consent is required based on the design of the study.

Having available previous data from 1983 to 2004 about mesothelioma deaths in Greece, MM mortality rate has been estimated for each year in a span from 1983 to 2015. Afterwards, this 33-year period was divided into three 11-year consecutive periods, namely 1983–1993, 1994–2004 and 2005–2015, in order to conduct comparative analysis.

In order to estimate the cause-specific mortality rate of Mesothelioma for each geographical area in Greece for the period 2004–2015 (12 years), we used the following Equation [16]:Mesothelioma mortality rate = (Total number of mesothelioma deaths2004–2015 in area/middle population in 2009 in area) × 100.000/12years.(1)

These new data were compared with previous published data related with mesothelioma mortality in Greece [17]. All data were analyzed with one-way ANOVA using PRISM 6.0 software. Data were assessed for normality with the D’Agostino & Pearson test and parametric analysis among groups was performed with the one-way ANOVA test followed by Tukey’s post-hoc test for multiple comparisons. Non-parametric analyses among groups were performed with Brown–Forsythe–Welch ANOVA tests followed by Dunnett’s T3 post-hoc test. A *p* value less than 0.05 was considered statistically significant.

## 3. Results

According to our data, currently MM deaths in males account for about 77.5% of cases, and the males: females ratio is 3:1 (Figure 1).

Having estimated the mesothelioma mortality rates (MMRs) for each consecutive period it was found that the MMR was significantly increased during the last period 2005–2015. This increase was sevenfold and a twofold compared to the mesothelioma mortality rate of the periods 1983–1993 and 1994–2004, respectively (Figure 2).

Similarly, a significant increase was observed in MMRs both in males (Figure 3A) and females (Figure 3B). More specifically, we found an 11-fold increase in MMR in males between the first and the third studied period (1983–1993 and 2005–2015), a fivefold increase in the first and the second studied period (1983–1993 and 1994–2004) and a twofold increase between 1994–2004 and 2005–2015. Similarly, we found an approximately fivefold increase in MMR in females between the first and the third studied period(1983–1993 and 2005–2015), a twofold increase between the first and the second period studied (1983–1993 and 1994–2004) and a twofold increase between 1994–2004 and 2005–2015. Comparing the last two periods, the twofold increase in MMRs for both genders is evident.

Figure 4 depicts the MM deaths by sex in every three-year period from 2004 to 2015. The curves are similar and gradually increase. Also, the curves show a peak during the period 2010–2012. As far as the male deaths plot is concerned, it is very close to the plot of total deaths. This is in accordance with the fact that MM inflicts mainly males, since it is considered an occupational disease.

Looking at the MM deaths by sex and age, we show that the maximum number of deaths has occurred in the age group 70–79 for both genders, during the period 2005–2015 (Figure 5A). This is a shift since the period 1983–2004, that the maximum number of deaths was observed in the age group 60–69 irrespective of gender (Figure 5B).

MMRs for each geographical area in Greece have been estimated for the period 2004–2015 (12 years). The highest rate was found in Sterea Ellada, followed by Epirus, Peloponnisos, Macedonia-Thraki, Crete and Thessalia (Figure 6). Comparing these data with those published from a previous period, 1983–2003, an increased MMR rate was observed, in almost all geographical areas in Greece [17]. Especially, MMRs have been significantly increased in all areas except for Epirus, which remained exactly the same as in 1983–2003.

## 4. Discussion

Our results showed that MMR in Greece has increased both in males and females and in almost all geographical areas during the period 2005–2015. Comparing with data from previous decades it was observed that this increase was sevenfold and twofold greater from the periods 1983–1993 and 1994–2004, respectively. This sevenfold increase from the first period (1983–1993) could be attributed to the improvement in diagnostic procedures during the last two decades and also, to improved maintenance of the statistical data by the Hellenic Statistical Authority. However, the twofold increase between the last two studied periods (1994–2004 and 2005–2015) could be related to the use of asbestos in Greece as well as the improvement in diagnostic procedures. The increased MMRs may at least in part correlate with increased incidence reports due to increased mesothelioma awareness and diagnosis. The increased MMR during the first decade of the 21st century could be explained by the apparent growing consumption of asbestos in Greece since the 1970 [18]. According to data from the US Bureau of Mines, the production of asbestos in Greece from 1930 to 1964 was sporadic and in order of magnitude yielded 2–67 tons/year [19]. The first asbestos production factory opened in 1961 in Nea Lampsako in Sterea Ellada and ceased operations in the early 1990s [18]. Also, the main asbestos deposit of Greece was in Western Macedonia and Epirus. The largest deposit of asbestos was in Zidani, part of the county of Kozani, located in Macedonia, where mining had begun in the 1970s [12]. Greece had four asbestos plants and a production capacity that reached 300.000 tons annually in 1995. Therefore, asbestos production started in Greece since the 1960s and flourished in 1970s onwards, with a peak production in 1996 [11]. Although since 1999 the production has ceased, asbestos exportation continued until 2003 [11]. Taking into consideration the long latency period of MM occurrence (30–40 years), the great increase in the mortality rate during 2005–2015 is not surprising. 

In 1973, the International Agency for Research on Cancer (IARC) gathered sufficient evidence that asbestos and all its commercial forms are human carcinogens [20]. USA banned the use of certain asbestos products in the early 1970s. In European countries the ban of asbestos use was introduced a decade later e.g. in Iceland in 1983, Norway in 1984, Denmark and Sweden in 1986, Netherlands in 1991, Italy and Finland in 1992, Germany and Croatia (only crocidolite and amosite) in 1993, France and Slovenia in 1996, Poland in 1997, UK in 1999, Latvia in 2000, Spain in 2002 and in Greece in 2005 [4].

It is well known that the current global burden of asbestos related cancers has been reflected in the asbestos usage in the 20th century [20]. Ιt was observed that the incidence of MM was decreasing in the countries where the ban of asbestos was applied early, contrary to the countries where the law on banning asbestos use was delayed [1]. Recent studies about MMRs have shown mixed trends. For countries with early and universal asbestos ban, like Sweden, and Finland the rates decline progressively. In contrast, for countries with late or incomplete asbestos ban, the rates are still increasing e.g. Japan, Croatia and Poland [1]. Montanaro et al., suggest that the earlier the asbestos ban in countries the more moderate increases in the incident of mesothelioma rates [1]. Boffetta et al., reported that in 2013 the overall mesothelioma rates among males tended to decrease in Netherlands, Australia, US, France and UK, but tended to increase in Japan and Poland [21]. The overall mesothelioma rates in females were between 0.1 and 0.5/100,000, and the highest one, in a span from 2000 to 2004, in Italy, UK and Australia. In contrast, an appreciable change in female mortality rate was observed between 2000-2004 and 2013 in Poland, whose rates rose from 0.07 to 0.21/100000 [11].

Another important issue is the shift of maximum number of mesothelioma deaths to older ages 70–79 years during the period 2005–2015, comparing with that during a span from 1983 to 2004. Deaths due to mesothelioma in younger ages (maximum number of deaths at ages 60–69 years) during the period 1983–2004 is mainly attributed to occupational asbestos exposure, which is continued in early adulthood. It is known that the production of asbestos in Greece occurred since 1960 and this industry flourished in 1970s onwards, with a peak in production in the mid-1990s. We have mentioned that asbestos mines in Greece closed in the 1990s [12]. Mesothelioma most commonly affects workers in asbestos mines and in places where asbestos is processed and transported. Other victims are construction workers or workers who handle asbestos tiles, insulation materials, and pipes covered in asbestos; those who repair brakes on older vehicles, electrical appliances such as old irons and hair dryers [18]. Twenty-three to 44 years have passed from the start of asbestos production in Greece (1960) until the first studied period (1983–2004). During this period, most of subjects that died from mesothelioma were 60–69 and 50–59 years old. Taking into consideration the age of starting work and the long latency period of mesothelioma (30–40 years), these results demonstrate occupational exposure. Recent study showed that mortality rates in older age groups was increased between 2000 to 2004 and 2013, whereas the rates in younger groups was decreased between the same period. The same pattern was identified among females [11].

The increase of MMR in nearly all geographical areas in Greece was expected because of the significant increase of total rate in Greece. The fourfold increased mortality rate in Sterea Ellada and Macedonia-Thraki prefectures during 2005–2015, compared to the period 1983–2004, is noteworthy, too. This increase is justifiable if we consider that the first factory of asbestos opened in Nea Lampsakos in Sterea Ellada, as well as the largest deposit of asbestos in Greece was found in Zidani in Macedonia [18,19]. The first factory operated from 1961 to early 1990s and the second from early 1970s to 1998. Thessalia is another prefecture of Greece where increased MMR was observed. A sixfold increase was found, compared to previous data (1983–2004), which could be attributed to improved diagnostic procedures in the Department of Respiratory Medicine of the University of Thessaly and the associated increased awareness. A closer look at the rates in the Greek prefectures in the two successive periods shows that the rate in Epirus prefecture has remained exactly the same. This is a very interesting finding that demonstrates the value of increased awareness and the implementation of the appropriate public health policies. Since 1969, in Metsovo, an area located in the prefecture of Epirus, a high prevalence of endemic calcification of the pleura was observed that was onwards called the “Metsovo Lung” [22,23]. The residents of Metsovo used an asbestos-contaminated soil (called “luto” in the local dialect) for whitewashing and applying to the interior walls. Bazas et al., and Konstantopoulos et al., have carried out studies and had proved the health hazards imposed by luto [22,23]. After that a health promotion strategy was implemented and luto has not been used since 1985, leading to the end of the domestic exposure epidemic [24,25]. For the time, the rate in Epirus is remaining the same and a decreased rate is anticipated in the following years. The increased surveillance in Epirus resulted in stabilizing of MMRs. Unfortunately, there are no available data for other Greek prefectures.

## 5. Conclusions

Our data show an increase of the MMR in Greece for the period 2005 to 2015 as compared to the two previous decades. This finding probably reflects the results of the production and use of asbestos in Greece that begun in 1960 and ended in the end of 1990s, since the latency period for MM is 30–40 years. Most probably the MM epidemic in Greece has not yet peaked. Further increases are expected in the next decades, since the ban of asbestos in Greece was implemented in 2005.The stable MMR in Epirus is an example that the awareness and public health promotion strategies are crucial in minimizing the burden of the disease.

## Figures and Tables

**Figure 1 medicina-55-00419-f001:**
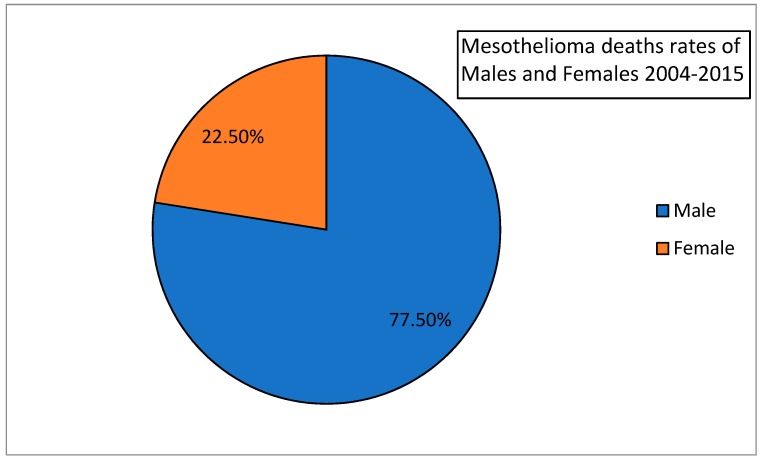
Death rates due to MM in Greece presented by gender for the period 2004–2015. Out of 455 total deaths, 353 were male patients.

**Figure 2 medicina-55-00419-f002:**
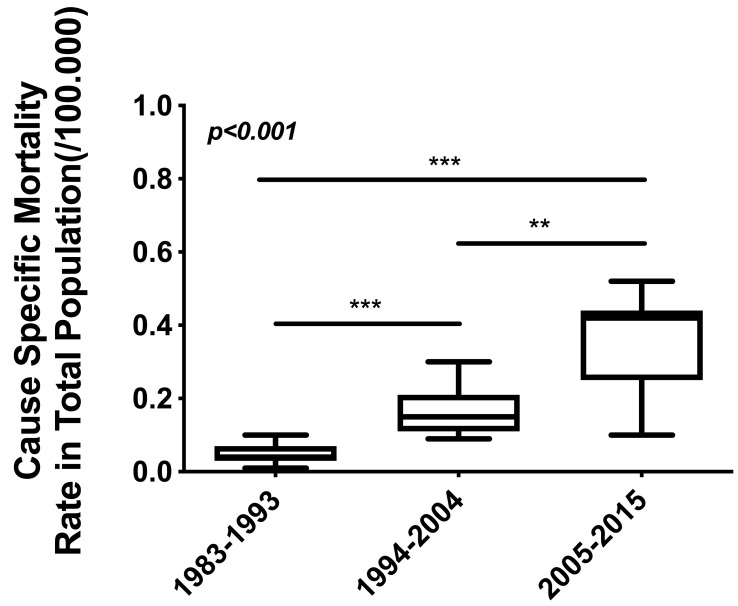
The mesothelioma mortality rate (MMR) in the Greek population for 3 consecutive periods. During the period 1983–1993 the MMR was 0.048/100,000 population, in 1994–2004 the MMR was 0.16/100,000 population and in 2005–2015 the MMR was 0.35/100,000 population. (ANOVA analysis: *p* < 0.001). ***p* < 0.01, ****p* < 0.001.

**Figure 3 medicina-55-00419-f003:**
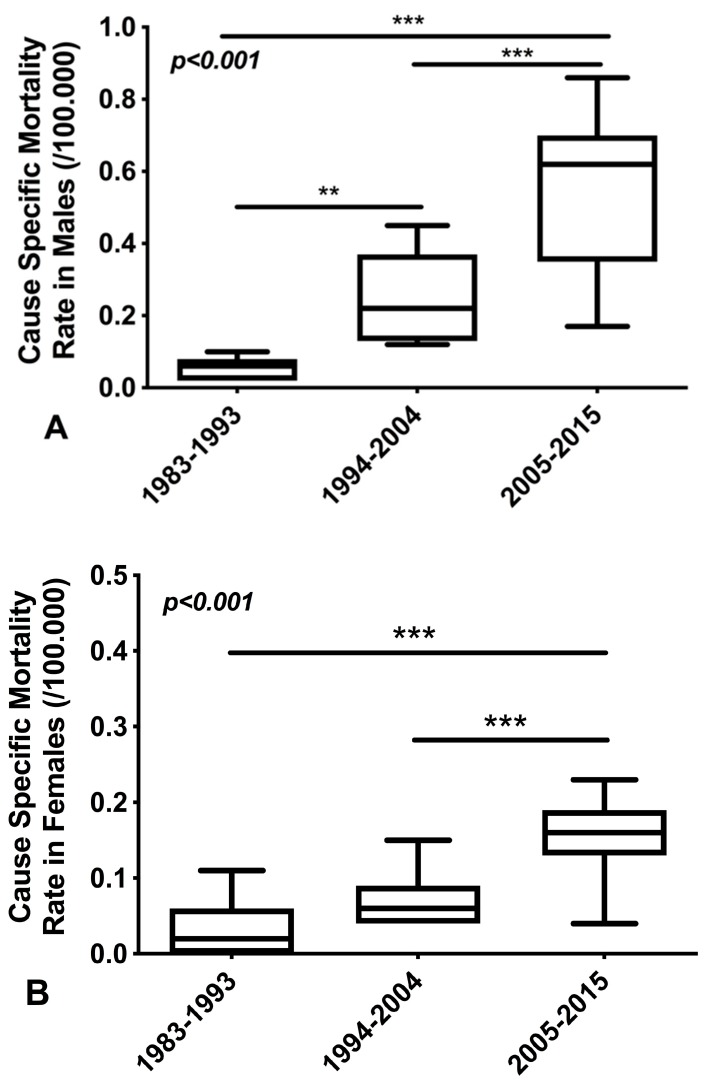
Greek MMR in Males(**A**) and in females (**B**)for the three consecutive periods. (**A**) Regarding males during 1983–1993 the MMR was 0.058/100,000, during 1994–2004 the MMR was 0.25/100,000 male population and during 2005–2015 the MMR was 0.56/100,000. (ANOVA analysis: *p* < 0.001). (**B**)Regarding females during 1983–1993 the MMR was 0.037/100,000, during 1994–2004 the MMR was 0.074/100,000 and during 2005–2015 the MMR was 0.16/100,000. (ANOVA analysis: *p* < 0.001).

**Figure 4 medicina-55-00419-f004:**
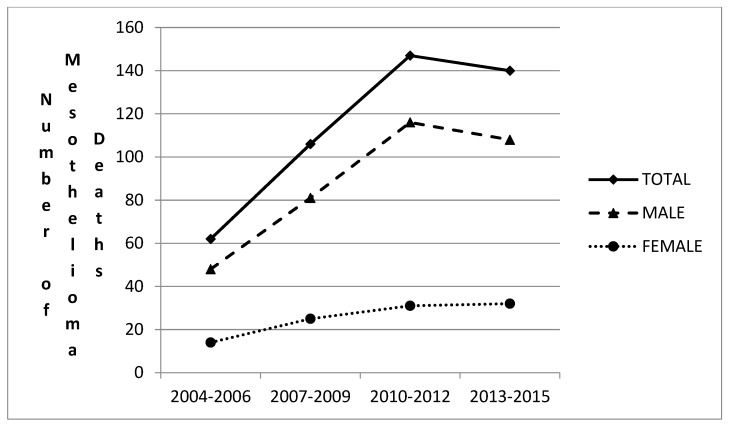
Deaths by gender and three-year intervals for the 2004–2015 period.

**Figure 5 medicina-55-00419-f005:**
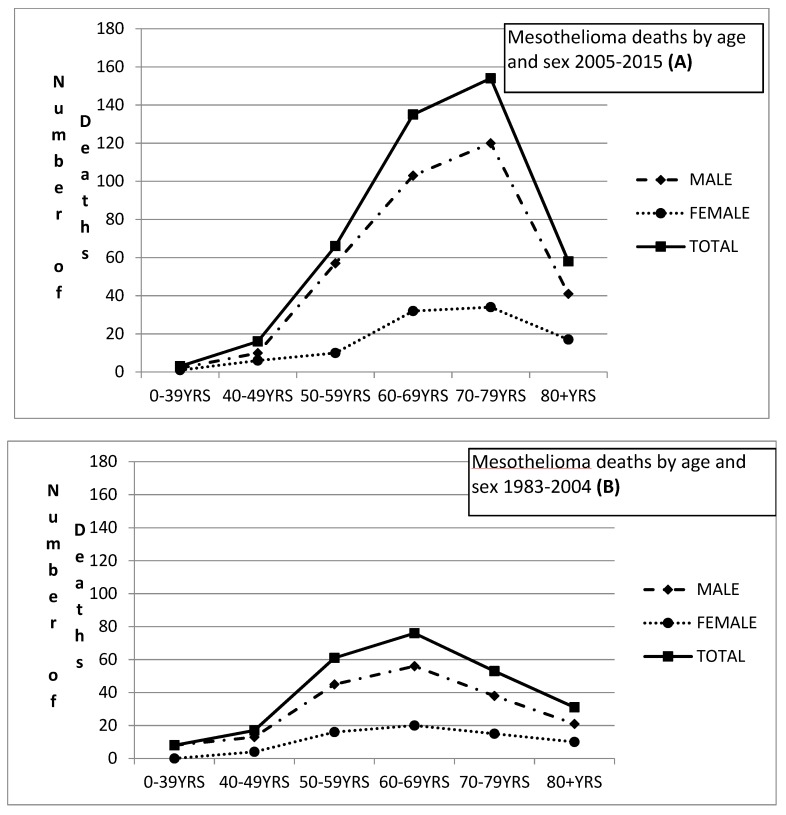
Mesothelioma deaths by age and sex during 2005–2015 (**A**)and 1983–2004 (**B**).

**Figure 6 medicina-55-00419-f006:**
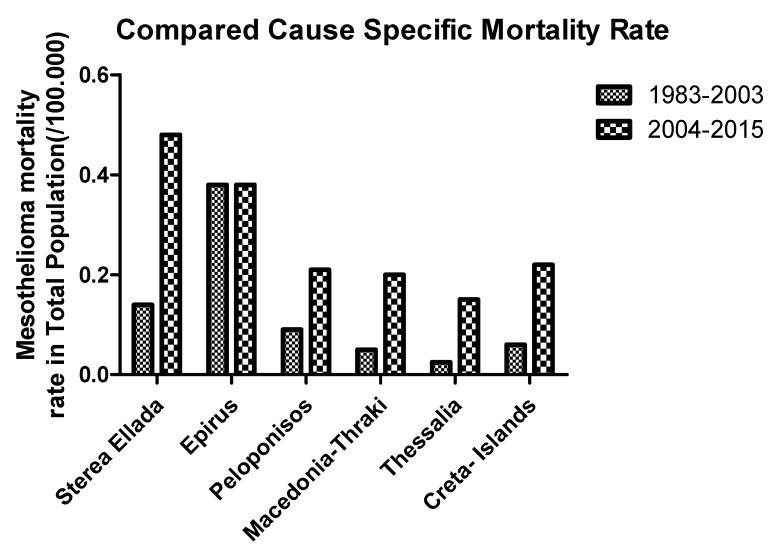
MMR in Greek geographical areas during two consecutive periods 1983–2003 and 2004–2015.

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
