# Peer review of "Mesothelioma Mortality Rates in Greece for the Period 2005–2015 Is Increased Compared to Previous Decades"

_medicina, 2019, doi:10.3390/medicina55080419_

Round 1

Reviewer 1 Report

The manuscript shows an increase of the mortality rate due to malignant mesothelioma (MM) in Greece for the period 2005-2015 as compared to the periods 1983-1993 and 1994-2004. Based on the well-known long latency of asbestos-induced carcinogenesis, the authors conclude that this mortality increase “probably reflects the results of the production and use of asbestos in Greece that begun in 1960 and ended in the end of 1990s”. This conclusion is fair, and the manuscript is fairly well written. However, some improvements seem necessary to make it fully publishable.

Introduction, line 36-37, “Exposure to asbestos is considered as the primary aetiologic agent of mesothelioma”. Given the epidemiological character of the manuscript, the authors should also mention other risk factors for/causes of MM, such as radiation, chronic serosal inflammation (fx TB), and asbestos-like fibers (erionite, fluoro-edenite, nonotubes). Examples that can be cited are the erionite-induced cases of MM in Cappadocian villages in Turkey or in the Western States of the USA, North Dakota and Montana as well as the cases of fluoro-edenite-induced MM in the Sicilian town of Biancavilla. Finally, the possibility of genetic predisposition, especially the so-called BAP1 cancer syndrome that is due to germline BAP1 mutations and has been associated with familar cases of MM in the above-mentioned Cappodacian villages.

Line 42-44, “The diagnosis can mainly be established by biopsy [2]. The “gold standard” for the diagnosis is thoracoscopic or after endoscopic biopsies where mesothelioma arise, for the confirmation of diagnosis [6]”. The two sentences are not so clear with respect to international guidelines for the diagnostics of MM and should probably be reformulated. The final diagnosis of MM (in the context of proper clinical, radiological, and thoracoscopic/surgical findings) is always histological (on biopsy or resection); in a few particular cases it can be made on cytological material (Husain AN et al. Guidelines for Pathologic Diagnosis of Malignant Mesothelioma 2017 Update of the Consensus Statement From the International Mesothelioma Interest Group. Arch Pathol Lab Med. 2018 Jan;142(1):89-108. doi: 10.5858/arpa.2017-0124-RA.)

Figure 2, 3 & 6: Can the authors provide P values proving that the differences are significant?

Results, line 118, “is accordance with the fact that mesothelioma inflicts mainly males”: it should be “is in accordance with the fact that mesothelioma affects mainly males”.

Discussion, line 147-8, “the 2-fold increase between the last two studied periods (1994-2004 and 2005-2015) could be related to the use of asbestos in Greece”. This makes sense but does not exclude that improved diagnostic procedures have also played a role in the period 2005-2015 as well. After all, some of the diagnostic guidelines have been changed between the 1994-2015.

Line 162, “USA banned the use of asbestos in the early 1970s” and line 171-2, “For countries with early and universal asbestos ban, like USA …”: It is not entirely correct, as in practical terms asbestos is not completely banned in the US.  The U.S. Environmental Protection Agency (EPA) in 1973 banned certain asbestos products for fireproofing and insulating purposes and in 1989, the EPA attempted to implement a full ban on the manufacturing, importation, processing and sale of asbestos-containing products, but in 1991 asbestos industry supporters challenged and overturned the ban in a landmark lawsuit.

Are the authors presenting data on all types of mesotheliomas, fx. from all serosal organs/localizations (peritoneal, pericardial etc..) or only for pleural MM? This should be clarified. And is it possible to correlate the changes in MM-related mortality to the three main histotypes (epitelioid, sarcomatoid and biphasic)?

Minor errors/typos should be amended: “started at 1960” on line 60; “the Cause-SpecificMortality” on line 78; “irrespectively at gender” on line 125; “which was remained” on line 135.

Reviewer 2 Report

Gogou et al, assess the mesothelioma mortality rate in Greece for the period between 2004 and 2015 and compare the rates observed to the mortality rates from the previous decades. As expected, the mortality rates for the current decade (2005 - 2015) was much higher than the previous decade (1994 -2004). Mesothelioma incidence has not yet peaked in Greece due to the later ban on the use and production of asbestos in 2005. With a latency period that ranges from 10 - 40 years the highest mortality rates are found in the 70 -79 age group for both males and females, as that cohort would have had the longest exposure and latency period for asbestos. While the findings followed the expected trend for mesothelioma incidence and mortality rates in an area with recent asbestos mining activity and use, it underscores the need for better monitoring and early detection methods to reduce mortality rates and improve survival.

Comments:

1) It is confusing when the authors alternate between referring to their study period as being 2004 to 2014, between 2005 to 2015 and between 2004 and 2015. Please pick one range to define and use for consistency. The authors later define the eleven year periods they compared which makes their study period clear. It would be helpful to see it defined the same way in the abstract as well as at the end of the introduction.

2) The authors do not specify whether the deaths attributed to mesothelioma where in patients with a confirmed mesothelioma diagnosis or whether mesothelioma as a cause of death was found at autopsy or a combination of both. The only prefecture they mentioned with increased surveillance is Epirus. The increased surveillance however, only resulted in stabilizing of mortality rates. The questions remaining are: is the same practice being implemented in other prefectures?  Do the increased mortality rates correlate with increased incidence reports due to increased awareness and diagnosis? How will this affect Mesothelioma incidence rates in the future?

3) Typographical errors: line 135 -  "...which was remained, exactly the same..." should read "...which remained exactly the same as in 1983 - 2003." 

Line 142 "...and almost in all geographical areas..." consider changing to "...in almost all geographical areas..." 

line 156 "... asbestos production had started in Greece since the 1960s..." consider "...asbestos production in Greece started in the 1960s..."

Line 158 " Although since 1999 the production has ceased, asbestos exportation was continuing..." rephrase sentence and be careful of tense agreement. First half of the sentence was in the past tense while the second half was in the present continuous tense.

Line 162 "it's" should be its...
